# PRE-TRAIN WITH BACKPROPAGATION AND FINE-TUNE WITH A BIO-PLAUSIBLE LEARNING RULE

## ABSTRACT

Backpropagation (BP) has long been the cornerstone of deep neural network training. While neural networks trained with backpropagation typically have high accuracy and precision, they suffer from limitations in their robustness to adversarial perturbation. Biologically plausible (bio-plausible) learning rules, on the other hand, are more robust. Yet, they typically underperform in terms of accuracy and precision, which has limited their widespread adoption. In this work, we aim to bridge this gap. We propose a novel approach where neural networks are pre-trained using backpropagation and fine-tuned using bio-plausible learning rules. We use several types of Sign-Symmetry learning methods to fine-tune models pre-trained using backpropagation. We explore the effectiveness of this approach in two tasks, image classification and image retrieval, then demonstrate that it improves robustness against gradient-based adversarial attacks while offering comparable accuracy and precision compared to the use of backpropagation alone. These findings show the benefit of mixing backpropagation and bio-plausible learning rules, suggesting the need for further research by the community to evaluate this approach on other tasks.

## 1 INTRODUCTION

The human cerebral cortex learns through updating synapses based on input signals that activate multiple regions involved in the learning process Hebb (2005); Markram et al. (1997); Bliss & Lømo (1973). The complex multilayered structure of our brain's neural network makes it challenging to determine the exact mechanism responsible for learning in the brain Lillicrap et al. (2020). A common way to address this problem in Deep Artificial Neural Networks is through a credit assignment algorithm responsible for updating synaptic connections based on feedback signals. These algorithms typically differ in the way the feedback signal is backpropagated Lansdell et al. (2019). The most widely used credit assignment algorithm is Backpropagation (BP) Rumelhart et al. (1986).

While BP has been an indispensable tool for building effective neural networks, studies have shown that it is not robust to perturbation attacks Szegedy (2013). Adversarial perturbation is a phenomenon commonly observed in neural networks, wherein carefully crafted inputs can deceive models into making incorrect predictions. An attacker can exploit the weaknesses of a trained neural network by introducing subtle perturbations to an input image that are not perceivable to the human eye but significantly alter the model's output. This alteration could be as simple as adding a small amount of noise to the image (e.g., slightly changing pixel values). Such modifications can result in the model's accuracy dropping dramatically, even though the altered inputs are nearly indistinguishable from legitimate inputs Zhang et al. (2019b).

Unlike backpropagation, bio-plausible learning rules are more robust to adversarial attacks. These learning rules generally refer to training methods for artificial neural networks that closely mimic the mechanisms observed in biological systems. In this approach, properties found in natural learning mechanisms are applied to artificial neural networks, aligning them with their natural counterparts. Examples of these learning rules include Feedback Alignment (FA) Lillicrap et al. (2014) and Sign-Symmetry rules, such as Fixed Random-magnitude Sign-concordant Feedback (frSF), Batchwise Random Magnitude Sign-concordant Feedbacks (brSF), and Uniform Sign-concordant Feedback (uSF) Liao et al. (2016).

While these learning rules are more robust to adversarial attacks, they typically underperform in terms of accuracy and precision when compared to backpropagation. Multiple studies have demonstrated that the performance disparity between BP and FA can be substantial. For example, Bartunov et al. (2018) showed that, on the task of image classification, the BP achieves a top-1 error rate reaching 71.43% which is much better than the 93.03% achieved by FA (on ImageNet). Despite the robustness of bio-plausible learning rules, their inferior performance compared to backpropagation has significantly limited their widespread adoption. In this work, we aim to bridge the gap between BP and bio-plausible methods. We propose to reconsider the integration of bio-plausible methods when training deep neural networks through backpropagation.

We suggest a novel approach where neural networks are pre-trained using backpropagation and fine-tuned using bio-plausible learning rules. In particular, we use the Sign-Symmetry learning rules to fine-tune models that were pre-trained using backpropagation. We explore the effectiveness of this approach in two tasks, image classification and hashing-based image retrieval, and show that neural networks trained using this method have significantly better robustness against gradient-based adversarial attacks while being close to backpropagation in terms of accuracy. The main intuition behind our contribution is the following: the bio-plausible learning algorithms that we consider do not rely on computing precise gradients to update weights; making them more resilient against gradient-based adversarial attacks.

We assess the effectiveness of our approach in different scenarios: 1) For different backbones pre-trained using backpropagation (Alexnet, VGG16 and ResNet-18); 2) For different bio-plausible learning rules (uSF, frSF, and brSF); 3) For different gradient-based adversarial attacks (2 attacks for each task); and 4) For different datasets (at least 3 datasets for each task). Our robustness results demonstrate that models fine-tuned using our method were more robust to gradient-based adversarial attacks in all the cases while still having performance that is close to classical backpropagation in terms of accuracy. These findings show the benefit of mixing backpropagation and bio-plausible learning rules, suggesting the need for further research by the community to evaluate this approach on other tasks.

One of the main motivations behind this work is to explore the effectiveness of bio-plausible learning methods in the task of fine-tuning (i.e., mapping high-quality embeddings to an output). Our findings suggest that sign-symmetry bio-plausible learning methods (which we study in this paper) can map high-quality embeddings to an output, assuming such embeddings are effectively created (in this work we create them using backpropagation). This suggests the need for more research on the development of bio-plausible learning methods that can effectively learn high-quality embeddings, which would enable the use of bio-plausible learning rules end-to-end for the effective training of neural networks.

It also highlights the need to evaluate bio-plausible learning methods using more metrics, such as the robustness against gradient-based attacks, in addition to the classical metrics used in the literature.

The contributions of this paper are as follow:

- We propose a novel method for training deep neural networks for the tasks of image classification and hash-based image retrieval: pre-train backbone models using backpropagation, and fine-tuning them using Sign-Symmetry learning rules.

- We demonstrate that the proposed approach leads to significantly better robustness to gradient-based adversarial attacks while still providing close accuracy compared to backpropagation alone.

- We open-source all of our code to the research community, opening the door for further research in this area.

## 2 RELATED WORK

**Backpropagation and Bio-plausible Learning Rules**  While known to be the most efficient learning algorithm, backpropagation has been criticized for being biologically implausible in many aspects, particularly the weight transport problem Grossberg (1987); Crick (1989); Schwartz (1993). It is believed that the brain uses asymmetrical feedback signals, whereas BP uses the same weights for both the forward and backward passes Lillicrap et al. (2020). Most promising work on bio-

logically plausible credit assignment methods builds upon backpropagation and attempts to minimize weight transport. Figure 1 illustrates learning methods relevant to this work and the degree to which weights are used to conduct the backward pass. The backpropagation algorithm uses a symmetrical feedback structure, representing the most extreme example of weight transport. Sign-Symmetry Liao et al. (2016) uses the sign matrix of the weights to conduct the feedback pass, with three common forms: Uniform Sign-Concordant Feedback (uSF), which uses the exact sign of the weight matrix, representing a less extreme example of the weight transport problem, Fixed Random Magnitude Sign-Concordant Feedback (frSF), and Batchwise Random Magnitude Sign-concordant Feedbacks (brSF), which are a type of approximate gradient that has been shown to be sufficient for learning and does not require temporally locked gradients Czarnecki et al. (2017). The raw form of the approximate gradient is known as Feedback Alignment (FA) Lillicrap et al. (2014), where a fixed random feedback matrix is used for learning, entirely avoiding weight transport. This class of methods, which uses a synthetic backward matrix, was supported by research Lillicrap et al. (2016) that claims symmetry is not necessary to train neural networks. Counter-intuitively, fixed random feedback matrices could force the feedforward synaptic connections to align with their weights by making the error derivative, calculated by the forward weight matrix, close to the synthetic one that is calculated by the synthetic backward matrix Lillicrap et al. (2020), hence the name 'feedback alignment'. Moreover, biologically plausible alternatives could potentially address BP's sequential nature that renders it computationally inefficient. Methods relying on local information to update synaptic connections, such as Target Propagation, have been proposed Le Cun (1986); Hinton et al. (2007); Bengio (2014); Lee et al. (2015), but they do not scale in performance compared to backpropagation Bartunov et al. (2018), discouraging further research on them.

**The Robustness of Bio-plausible Learning Rules** When it comes to assessing the robustness of bio-plausible learning rules, relatively little work has been conducted in this area. In the study by Sanfiz & Akrout (2021), the authors evaluated the robustness of bio-plausible credit assignment methods, including FA, uSF, frSF, and others. Their findings demonstrated that these methods exhibit enhanced performance under certain conditions. The study employed various attack methods, including white-box (gradient-based) attacks such as FGSM Goodfellow (2014), PGD Madry (2017), APGD Zimmermann (2019), and TPGD Zhang et al. (2019a). For black-box attacks, the researchers utilized the Few-pixel attack Su et al. (2019) and Square attack Andriushchenko et al. (2020). Results from the white-box attacks were particularly useful in highlighting the differences in robustness between backpropagation (BP) and bio-plausible learning rules. However, black-box attacks yielded similar results across the methods and did not provide sufficient evidence to claim any general tendencies in robustness.

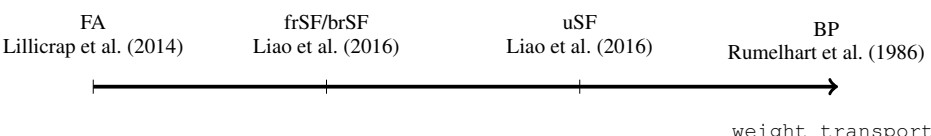

Figure 1: Credit assignment methods can be ranked based on the amount of weight transport each method employs. Backpropagation (BP) is known to use full weight transport, as the algorithm utilizes the same forward weight matrix to conduct the backward pass. Uniform Sign-concordant Feedback (uSF) uses the sign of the weight matrix as a backward matrix, thus utilizing less information than BP. Further Fixed Random Magnitude Sign-Concordant Feedback (frSF) and Batchwise Random Magnitude Sign-concordant Feedbacks (brSF) add approximation to the sign of the weight matrix, which introduces more randomness and uses less weight information. Finally, Feedback Alignment (FA) uses no weight transport, relying instead on a fixed random matrix to conduct the backward pass.

## 3 BACKGROUND

### 3.1 LEARNING ALGORITHMS

In the following, we consider a fully connected neural network with $L$ layers. $W_l$ is the weight matrix for layer $l$. $\mathbf{a_l}$ denotes the pre-activation of layer $l$, and satisfies $\mathbf{h_l} = \mathbf{f}(\mathbf{a_l})$, where $\mathbf{h_l}$ is the activation vector for layer $l$ and $\mathbf{f}$ is a non-linearity. The network final output is denoted $\hat{y}$. We calculate the error using the squared error $E = \frac{1}{2} \sum_k (y_k - \hat{y}_k)^2$.

**The Backpropagation algorithm**  In backpropagation, we calculate the exact error for each parameter of the network and update its value using the negative of its gradient, given in vector/matrix notation: Lillicrap et al. (2020):

$$\Delta W_l = -\eta \frac{\partial E}{\partial W_l} = -\eta \boldsymbol{\delta}_l \mathbf{h}_{l-1}^\top \tag{1}$$

Where $\eta$ is the learning rate and $\delta$ is referred to as the error signal and is computed recursively via the chain rule:

$$\boldsymbol{\delta}_l = \left(W_{l+1}^\top \boldsymbol{\delta}_{l+1}\right) \circ \mathbf{f}'\left(\mathbf{a}_l\right). \tag{2}$$

Where $\circ$ is the Hadamard product. This equation demonstrates the full symmetry of backpropagation, as the full weight matrix is used to calculate the backward update.

**Feedback Alignment**  In Feedback Alignment Lillicrap et al. (2014), the update of the synaptic connections is similar to BP given by equation 1, but instead of using the weight matrix to perform the backward update, a synthetic fixed random matrix is defined for each layer which we denote $B_l$. The error signal error in FA is then given by:

$$\boldsymbol{\delta}_l = \left(B_{l+1} \boldsymbol{\delta}_{l+1}\right) \circ \mathbf{f}'\left(\mathbf{a}_l\right). \tag{3}$$

FA introduces an unconventional method of updating synaptic connections compared to BP and employs a much simpler approach to circumvent the weight transport problem. This algorithm has been criticized for its lower performance compared to BP Bartunov et al. (2018). Sign-Symmetry methods try to mitigate this problem and therefore show more promising results.

**Sign-Symmetry**  The Sign-Symmetry algorithm Liao et al. (2016) shares similarities with Feedback Alignment in that it tries to mitigate the biological problem of weight transport and introduces symmetrical sign-sharing in feedback (instead of a random matrix, it uses the sign matrix of the feedforward weights). This algorithm strikes a balance between weight transport considerations and biological plausibility, leveraging benefits from both principles' benefits. Let's denote by $V$ the feedback weight matrix used in Sign-Symmetry. In backpropagation, the backward pass is done using the matrix of weights, i.e. $V = W^T$. In Sign-Symmetry, $V$ is defined through multiple variants of *asymmetric backpropagation*, while Liao et al. have explored various choices of asymmetric feedback, we only introduce the ones that are relevant to our work:

1. Uniform Sign-concordant Feedback (uSF): $V = \text{sign}(W^\top)$
2. Fixed Random-magnitude Sign-concordant Feedback (frSF): $V = M \circ \text{sign}(W^\top)$, where $M$ is defined as a matrix of uniform random numbers $\in [0, 1]$ and is initialized once and fixed through the training.
3. Batchwise Random Magnitude Sign-concordant Feedbacks (brSF): This is a variation of frSF where the magnitude matrix $M$ is redrawn after each parameter update.

### 3.2 ADVERSERIAL ROBUSTNESS

Adversarial robustness refers to a machine learning model's ability to maintain performance when faced with adversarial examples. There are two main types of adversarial attacks: white-box attacks, where the attacker has full knowledge of the model, and black-box attacks, where the attacker has limited information. In the following, we denote the original image as $x$, while $x^{\text{adv}}$ represents the adversarial example generated by the attack, and $y^*$ is its label. $f$ denotes the attacked model.

**Fast Gradient Sign Method** In FGSM Goodfellow (2014), the update to the image is given by:

$$x^{\text{adv}} = x + \varepsilon \, \text{sign} \left( \nabla_x \mathcal{L} \left( f(x), y^* \right) \right) \tag{4}$$

This attack is a one-step method where the image update is proportional to the sign of the gradient with respect to the original image $x$.

**Projected Gradient Descent** PGD Madry (2017) is an iterative variant of FGSM. It requires a step size $\alpha$ and a number of steps $k$ as hyperparameters. The update is given by:

$$x_0^{\text{adv}} = x_0, \quad x_t^{\text{adv}} = \text{Clip}_x^{\varepsilon} \left( x_{t-1}^{\text{adv}} + \alpha \, \text{sign} \left( \nabla_{x_{t-1}^{\text{adv}}} \mathcal{L} \left( f \left( x_{t-1}^{\text{adv}} \right) y^* \right) \right) \right) \tag{5}$$

where $\text{Clip}_x^{\varepsilon} \left( \cdot \right)$ is a function that ensures the input values remain within the $\varepsilon$-ball centered on the original image $x$.

**Hash Adversary Generation** HAG Yang et al. (2018) is a technique for generating adversarial examples to deceive deep hashing retrieval systems. It employs an iterative optimization process to modify query images, maximizing the Hamming distance between the original and adversarial hash codes. To overcome the challenge of discrete binary hash codes, HAG utilizes a surrogate gradient method. The technique primarily focuses on untargeted attacks, which do not aim at specific incorrect images. Instead, it seeks to confuse retrieval systems by producing semantically irrelevant results.

**Smart Deep Hashing Attack** SDHA Lu et al. (2021) is an advanced adversarial attack method that targets deep hashing models. Unlike HAG, SDHA adopts a more refined approach by using a dimension-wise Hamming distance strategy to generate adversarial examples. SDHA improves upon previous attacks by incorporating a ranking-sensitive approach. Instead of treating all dimensions of the hash code equally, SDHA prioritizes dimensions that are more vulnerable to perturbation. This strategy minimizes unnecessary changes to the image while maximizing the attack's impact on the retrieval results. SDHA also factors in the role of relevant images during adversarial example generation. By considering how similar images influence the average precision (AP) of the retrieval system, SDHA more effectively degrades the system's performance. An AP-oriented weight function is used to assign different importance to retrieved images based on their rank, enabling SDHA to target those contributing the most to the system's overall retrieval performance.

## 4 METHOD

**Tasks** We apply our proposed training method to two tasks: image classification, and hashing-based image retrieval. Since the first is well known, we will provide more details about the second. The task of hashing-based image retrieval aims to map high-dimensional image data into compact binary codes while preserving semantic similarity between images Hussain et al. (2022); Xia et al. (2014). The core idea is to transform images into low-dimensional binary hash codes that can be efficiently stored and quickly compared. The hashing-based image retrieval task involves two main steps. In the first, a hash function is learned. The goal of the hash function is to map input images to continuous encoding while ensuring that the assigned codes are semantically relevant Luo et al. (2022). The following step is the binary code generation: the hidden representation of the input image is binarized using a sign function Yang et al. (2022). During the retrieval process, the binary codes of query images are compared with those of database images using Hamming distance. This allows for rapid identification of potential matches Fang & Liu (2021).

**Learning Method and Model Architectures** We propose to use backpropagation for pre-training followed by the use of a Sign-Symmetry credit assignment method for fine-tuning. We use a CNN-based backbone model that was pre-trained on ImageNet (`IMAGENET1K_V1`) using backpropagation (e.g., RestNet-18). We then append a task-specific fully connected layer to adapt the network to the particular task, either a hash layer or a classification layer, with its parameters trained from scratch. Subsequently, we fine-tune all network parameters using a Sign-Symmetry method (e.g., uSF). We do not freeze the weights of the backbone during fine-tuning. We use the hashing loss HyP$^2$ Xu et al. (2022) to fine-tune the task of image retrieval, setting the hashing size $k$ to 32, while we use the Cross-Entropy loss function for image classification.

## 5 EXPERIMENTAL SETUP

**Models** We evaluate our method using three different backbone architectures: AlexNet Krizhevsky et al. (2012), ResNet-18 He et al. (2016), and VGG-16 Simonyan & Zisserman (2014). These pre-trained backbone models were obtained from the torchvision model repository.

**Datasets** We evaluate our approach on datasets of varying complexity, ranging from simple to highly complex, adhering to the same benchmarks and experimental settings established in the literature for each task. For image classification, we follow the experimental protocol outlined by Vuyyuru et al. (2020) with minor modifications. We utilize three datasets: CIFAR-10 Krizhevsky et al. (2009), ImageNet100 (a subset of ImageNet Deng et al. (2009) comprising 100 randomly selected classes), and the full ImageNet dataset with 1000 classes. For adversarial attacks, we follow the methodology proposed by Vuyyuru et al. (2020), where we conduct experiments using a 5,000-image test set for each dataset.

In hashing-based image retrieval, we adhere to the protocols established in the deep hashing litera-ture Schwengber et al. (2023); Cao et al. (2018); Zhu et al. (2016); Xu et al. (2022); Berriche et al. (2024). We use four datasets: CIFAR-10, NUS-WIDE, MS-COCO, and ImageNet100. CIFAR-10 is partitioned into 500 images per class for training, with 100 each for testing and validation. NUS-WIDE Chua et al. (2009), originally containing 269,648 images with 5,018 tags, is filtered to 148,332 images based on 21 prevalent concepts, with 10,500 for training and 2,100 each for testing and validation. MS-COCO Lin et al. (2014)'s 2017 release allocates 10,000 images for training and 5,000 each for testing and validation. ImageNet-100, a subset of ImageNet, comprises 100 classes with 13,000 training images and 5,000 evenly split between testing and validation. In all cases, the remaining images serve as the database for querying.

**Training** Pre-trained backbones, initially trained on ImageNet, were loaded through torchvision. For fine-tuning in both tasks, the same set of hyperparameters was employed. We used ADAM as the optimizer with $\beta_1 = 0.9$ and $\beta_2 = 0.999$, and a weight decay of 0.0005. The learning rate was set to $10^{-5}$ for all layers except the classification/hashing layer, which was assigned a learning rate of $10^{-4}$. We fine-tuned each dataset for 20 epochs using a batch size of 32.

**Evaluation Metrics** We assess performance using two metrics: accuracy for the classification task and mean average precision (mAP) for the hashing-based image retrieval task, which are both widely used for the tasks we consider Singh & Gupta (2022); Berriche et al. (2024); Bartunov et al. (2018). For the task of image retrieval, we follow the practices established in the literature Cao et al. (2017); Schwengber et al. (2023); Berriche et al. (2024) and compute the `mAP@k` with $k = 5000$ for CIFAR-10, NUS-WIDE, and MS COCO datasets, and with $k = 1000$ for ImageNet (details on how to compute mAP@k in Appendix A).

**Adverserial Robustness Evaluation Setup** Following the literature on adversarial robust-ness Athalye et al. (2018); Carlini et al. (2019); Uesato et al. (2018); Yuan et al. (2023) and studies on the robustness of bio-inspired methods Vuyyuru et al. (2020); Sanfiz & Akrout (2021), we use the same experimental setup with a few adjustments, which we highlight. Our implementation of adversarial attacks is based on the Foolbox Python package Rauber et al. (2017) for the classification task and on the work of Yuan et al. (2023) for the hashing-based retrieval task. Following the work by Vuyyuru et al. (2020), we employ the $L_\infty$ variants of FGSM and PGD to assess the robustness of classification models. FGSM is a single-iteration method that does not require any hyperparame-ters. For PGD, we set the step size $\alpha$ to $\varepsilon/3$ and executed the algorithm for 5 iterations. We vary the perturbation magnitude $\varepsilon$ from 0 to 0.5 and measure the robust accuracy.

We assess the robustness of the hashing-based image retrieval models against two non-targeted ad-versarial attacks: HAG and SDHA. These are iterative methods, for which we set the number of iterations to 5. We vary the perturbation magnitude $\varepsilon$ from 0.001 to 0.5. Through this configura-tion setup, we aim to compare the robustness behavior of each learning method as we increase the perturbation magnitude. Our primary focus is not on the attacks' effectiveness but rather on the relative performance of the hashing models under varying degrees of adversarial perturbation. This explains our decision to set the number of iterations to a relatively low value (5). This choice offers advantageous computational efficiency, although it may not converge to optimal results.

FGSM, PGD, HAG and SDHA are gradient-based attacks, meaning they rely on knowledge of the gradient to craft adversarial images. In our method, we propose using biologically plausible methods that possess the property of having approximate gradients, which makes it challenging to fool bio-trained models with such attacks.

# 6 RESULTS

We compared two approaches of fine-tuning: our proposed approach and the classical fine-tuning approach where backpropagation is used to fine-tune all the parameters of the model. We assess different variants of the models (using multiple backbones) using different datasets. We compare the performance and robustness of the models in each case (performance is measured using the accuracy for image classification and the mean average precision for image retrieval).

## 6.1 PERFORMANCE

**Image Classification** We first measure the accuracy in the classification task. Table 1 shows the accuracy for the 4 credit assignment methods that we use for fine-tuning (BP, frSF, brSF, and uSF), benchmarked on 3 datasets (CIFAR10, ImageNet100 and ImageNet) and when using one of the following backbones (AlexNet, VGG-16 or Resnet-18). The accuracy values presented in the table indicate that Sign-Symmetry methods can perform comparably to or outperform BP in certain backbone-dataset combinations, such as AlexNet-CIFAR10, AlexNet-ImageNet, and ResNet18-CIFAR10. Generally, uSF and frSF are the two methods that show the best results in classification.

Table 1: Accuracy measurements for credit assignment methods across various backbones and datasets. Bold values indicate the highest accuracy for each backbone-dataset combination. Underlined values denote the second-highest accuracy or two equally highest values. If three or more methods share the highest accuracy, no values are highlighted. Similarly, if two or more methods share the second-highest accuracy, none are underlined.

|  | Alexnet | | | VGG16 | | | ResNet-18 | | |
|---|---|---|---|---|---|---|---|---|---|
|  | CIFAR10 | ImageNet100 | ImageNet | CIFAR10 | ImageNet100 | ImageNet | CIFAR10 | ImageNet100 | ImageNet |
| BP | 90.62 | 100.0 | 84.37 | 100.0 | 100.0 | **100.0** | 93.75 | 90.62 | 84.37 |
| uSF | 90.62 | 100.0 | 84.37 | 100.0 | 100.0 | 96.87 | **96.87** | 87.5 | **87.5** |
| frSF | **93.75** | 100.0 | **90.62** | 100.0 | 100.0 | 93.75 | 90.62 | 84.37 | 84.37 |
| brSF | 87.5 | 100.0 | 84.37 | 90.62 | 100.0 | 87.5 | 90.62 | 90.62 | 84.37 |

**Hashing-based Image Retrieval** We asses hashing-based image retrieval models fine-tuned using one the 4 following credit assignment methods (BP, frSF, and uSF), on four datasets (CIFAR10, MS-COCO, NUSWIDE, and ImageNet) and when using one of the following backbones (AlexNet, VGG-16 or Resnet-18). Table 2 presents the results. Our findings indicate that Sign-Symmetry methods perform comparably to BP across most settings. Notably, uSF and frSF outperformed BP in 6 out of 12 benchmarks. In the remaining cases where BP demonstrated superior performance, uSF and frSF consistently achieved results close to BP, highlighting the stability and consistency of these methods. A particularly promising result was observed on the ImageNet dataset, which is widely recognized in the image retrieval literature as challenging due to its diversity and complexity. In this context, Sign-Symmetry methods outperformed BP in 2 out of 3 cases. These results suggest that Sign-Symmetry methods, particularly uSF and frSF, offer competitive performance and in some cases outperform BP.

## 6.2 ADVERSERIAL ROBUSTNESS

**Image Classification** We evaluate the robustness of the image classification models using two gradient-based attacks: FGSM and PGD. We benchmarked the accuracy of the fine-tuned models on three datasets (CIFAR10, ImageNet100, and ImageNet), using two backbone architectures (AlexNet and VGG-16). For each backbone:dataset configuration, we compared Sign-Symmetry methods to backpropagation and recorded the robust accuracy for each perturbation distance $\varepsilon \in \{0, 0.001, 0.005, 0.01, 0.05, 0.1, 0.5\}$.

Table 2: Mean Average Precision (mAP) values recorded for each benchmark dataset. For all the datasets used for the hashing task, $k = 5000$ except for ImageNet, where $k = 1000$. The HyP$^2$ loss function was utilized, and results are provided for three different backbone architectures: VGG16, AlexNet, and ResNet-18. The highest the better, in bold are the top value and underlined values are the second best.

| | Alexnet | | | | VGG16 | | | | ResNet-18 | | | |
|---|---|---|---|---|---|---|---|---|---|---|---|---|
| | CIFAR10 | MSCOCO | NUS WIDE | ImageNet100 | CIFAR10 | MSCOCO | NUS WIDE | ImageNet100 | CIFAR10 | MSCOCO | NUS WIDE | ImageNet100 |
| BP | **81.1** | **74.1** | 82.5 | 56.9 | **85.2** | 82.0 | 85.5 | 77.5 | **80.4** | 74.1 | **84.2** | **67.9** |
| frSF | 79.0 | 71.5 | **82.7** | 58.0 | 85.0 | 82.7 | 86.2 | 79.6 | 75.6 | 74.4 | 83.6 | 64.8 |
| uSF | 79.2 | 71.4 | 82.6 | **58.6** | 85.1 | **82.8** | **86.4** | **80.6** | 77.2 | 71.6 | 83.8 | 57.1 |

Figure 2 shows the results of the FGSM attack on multiple backbone:dataset configurations. A general trend is observed where Sign-Symmetry methods demonstrate better robustness compared to BP across all settings. This difference in robustness is sometimes substantial, as shown in figures 2b and 2e, where the gap reaches up to 21.87% at the maximum perturbation size $\varepsilon = 0.5$ and 65.62% when $\varepsilon = 0.05$. Among the Sign-Symmetry methods, brSF and frSF generally perform the best, with frSF exhibiting the most consistent performance across all experimental settings.

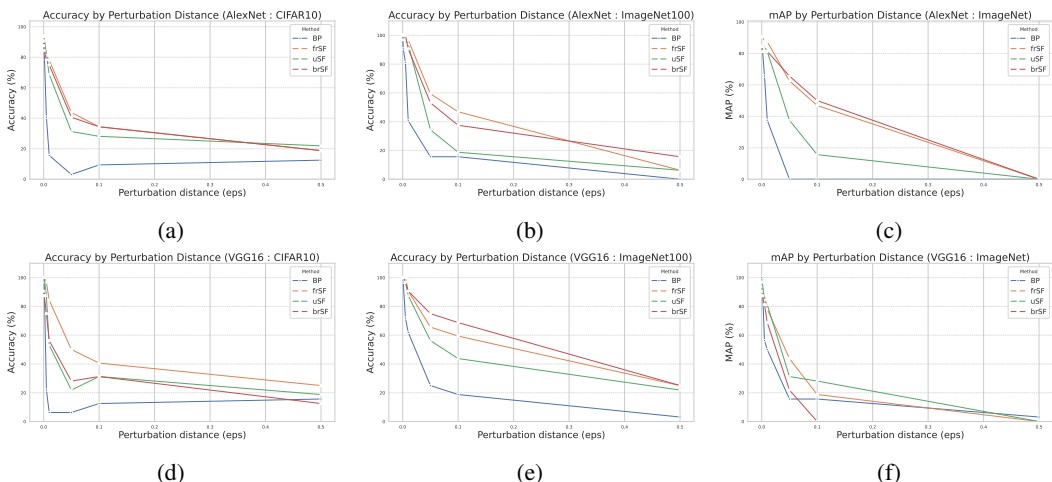

| (a) | (b) | (c) |
|---|---|---|
| (d) | (e) | (f) |

Figure 2: Evaluation of the FGSM adversarial attack on deep hashing models, showing classification accuracy as a function of perturbation magnitude $\varepsilon$. Results are presented for AlexNet (top row) and VGG-16 (bottom row).

Figure 3 presents the results for the PGD attack. Similarly, we observe that BP accuracy decreases drastically compared to Sign-Symmetry methods, which exhibit a more gradual decline. In experiments shown in figures 3a and 3d, BP's accuracy drops to 0 rapidly at $\varepsilon = 0.05$ and $\varepsilon = 0.1$, respectively. The difference in accuracy between BP and Sign-Symmetry methods can reach up to 25.0% when $\varepsilon = 0.5$ and 71.88% when $\varepsilon = 0.1$. Comparing the Sign-Symmetry methods under PGD attack reveals a trend similar to that observed with FGSM. frSF and brSF consistently outperform other methods, with frSF demonstrating the highest stability across various experimental conditions.

**Hashing-based Image Retrieval** We have measured the performance of deep hashing models when fine-tuned using either BP or Sign-Symmetry methods (frSF, brSF, uSF). The mean average precision was measured while varying the perturbation distance $\varepsilon$ from 0 to 0.5. This was done for both HAG and SDHA attacks, and for each experiment combination, as shown in Figures 4 and 5.

Figure 4 presents experiments conducted using the HAG attack, with subfigures showing results for different configurations using two backbones (AlexNet and VGG-16) and three datasets (CIFAR10, MSCOCO, and ImageNet). In Figure 4a, we observe that BP performance decreases significantly as the perturbation noise $\varepsilon$ increases, while the Sign-Symmetry fine-tuned model demonstrates robustness to increased perturbation. This trend is also evident in other experiments except for the AlexNet:MSCOCO configuration shown in Figure 4b, where the BP fine-tuned model exhibits solid

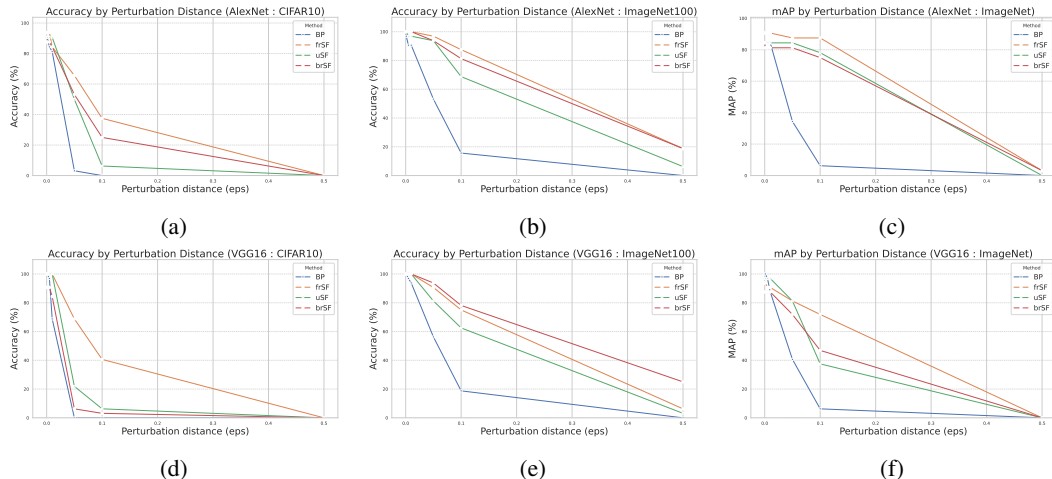

Figure 3: Evaluation of the PGD adversarial attack on deep hashing models, showing classification accuracy as a function of perturbation magnitude $\varepsilon$. Results are presented for AlexNet (top row) and VGG-16 (bottom row).

robustness but still remains below models fine-tuned using bio-plausible methods. In these experiments, the difference in robustness measures between BP and Sign-Symmetry methods reached up to 23.09% in the AlexNet:CIFAR10 configuration and up to 28.65% in VGG16:CIFAR10.

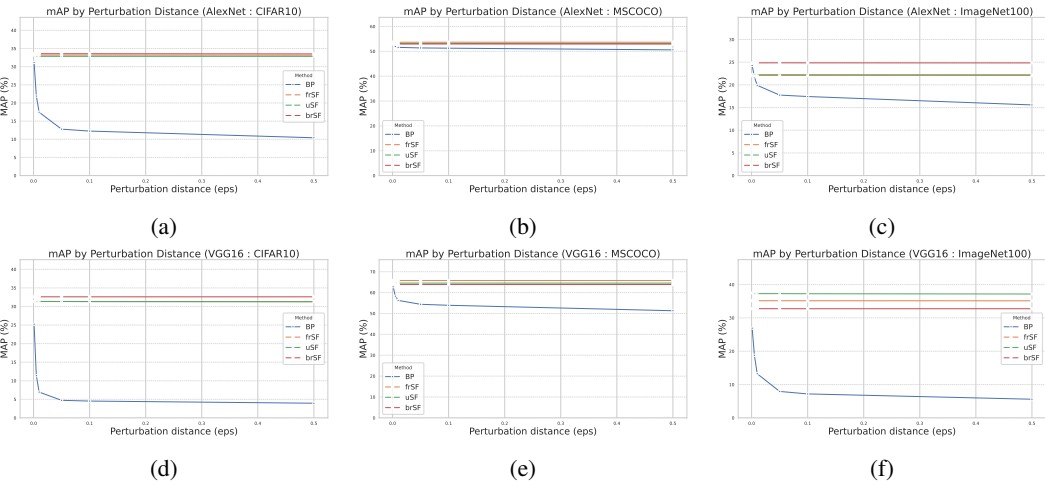

Figure 4: Evaluation of HAG adversarial attack on deep hashing models, measuring mAP@5000 as a function of perturbation magnitude $\varepsilon$.

Figure 5 presents the results of the SDHA attack on the deep hashing methods. The trend continues for BP, where we observe low robustness compared to Sign-Symmetry methods across all experiments. Figures 5a, 5b, 5d and 5e show that frSF is the most robust, followed by brSF and then uSF. In this experiment, the difference in robustness between BP and frSF reached up to 22.19% in VGG16:MSCOCO and up to 13.83% in AlexNet:CIFAR10.

## 7 DISCUSSION

Research by Sanfiz & Akrout (2021) has already highlighted the fact that bio-inspired learning methods are robust to adversarial attacks. This robustness is attributed to the use of approximate gradients during backpropagation, making adversarial attacks for gradient-based attacks more challenging. While previous work by Bartunov et al. (2018) has demonstrated the performance limitations of bio-plausible methods compared to backpropagation, no study has yet considered the use

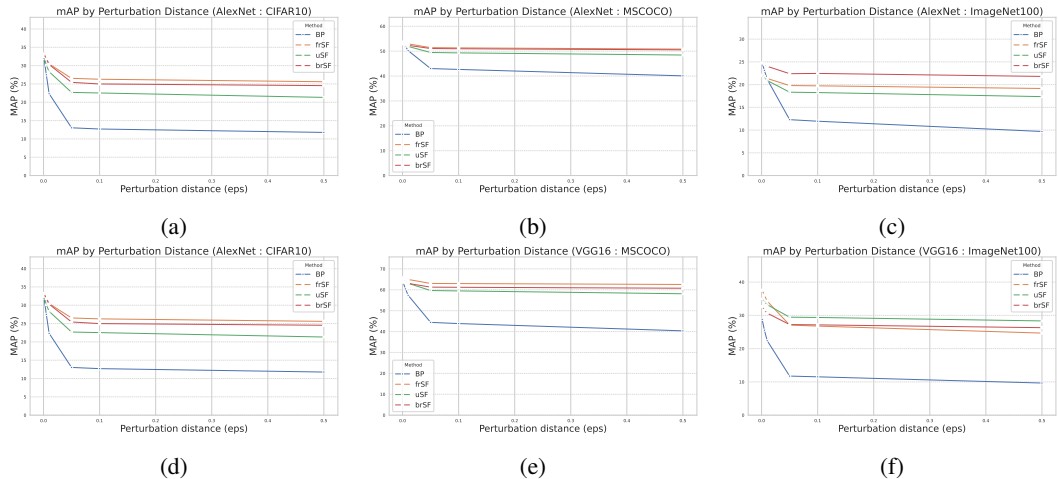

Figure 5: Evaluation of SDHA adversarial attack on deep hashing models, measuring mAP@5000 as a function of perturbation magnitude $\varepsilon$.

of bio-plausible methods along with backpropagation, potentially benefiting from both the effective learning of BP and the robustness of bio-inspired methods. Our research demonstrates that through fine-tuning pre-trained models using bio-plausible methods, we obtain models that achieve a performance comparable to BP while being more robust. Our findings also indicate that Sign-Symmetry methods are very effective in terms of performance and offer the greatest potential for results comparable to BP when used in fine-tuning. Among Sign-Symmetry methods, frSF emerged as the most performant and stable learning method. While our research has shown the effectiveness of the proposed approach, it is also important to investigate the direct impact of bio-plausible learning on the robustness of BP-trained methods. We suggest the use of a few fine-tuning steps using a Sign-Symmetry method as a corrective measure for robustness in BP-trained models. To fully explore this approach, more experiments need to be conducted with appropriate settings across various architectures. It should be noted that adversarial attacks are generally designed to target models trained using BP, especially white-box attacks that rely on the model's gradient. Given this, it would be more equitable to compare BP's robustness with attacks specifically designed for bio-plausible learning methods. This observation opens another avenue of research: developing attacks tailored to bio-plausible methods.

## 8 CONCLUSION

In conclusion, our proposed approach narrows the gap between bio-plausible learning methods and backpropagation. By integrating backpropagation with Sign-Symmetry methods, we have demonstrated the potential of achieving high robustness while maintaining performance comparable to BP. Improvements in robustness against adversarial attacks were significant and this was achieved on both tasks, image classification and hashing-based image retrieval. These results hold important implications for the field of deep learning, suggesting that biologically inspired learning rules can address some of the limitations of backpropagation, particularly in terms of adversarial robustness, without a significant degradation in performance. The observed improvements across various architectures, including AlexNet, VGG-16, and ResNet-18, and datasets of increasing complexity, such as CIFAR-10, ImageNet-100, ImageNet-1000, MS-COCO, and NUS-WIDE, further support the broad applicability and efficacy of this approach. Future research could investigate the application of this hybrid learning approach to a wider range of neural network architectures and tasks. Moreover, understanding the underlying mechanisms that contribute to the increased robustness observed in Sign-Symmetry methods could lead to the development of new biologically inspired learning rules that further enhance both performance and robustness.

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

## A    DETAILS ON THE EVALUATION METRICS

The mAP metric is defined as follows:

$$\texttt{mAP@k} = \frac{1}{|Q|} \sum_{q \in Q} AP_k(q) \qquad (6)$$

where $Q$ represents the set of queries, and $AP_k$ denotes the average precision of the first $k \leq n$ retrieved entries.

