# OpenReview forum: "PRE-TRAIN WITH BACKPROPAGATION AND FINE-TUNE  WITH A BIO-PLAUSIBLE LEARNING RULE"
_ICLR.cc/2025/Conference — Submitted to ICLR 2025_

### Official Review · Reviewer_Xapq · 2024-11-03

**Soundness:** 2
**Presentation:** 3
**Contribution:** 2
**Rating:** 3
**Confidence:** 3

**Summary:**

This paper is an attempt to propose a biologically plausible alternative solution to backpropagation that has been dominating the deep learning training for a few decades. Although biologically plausible learning methods are more robust to perturbation and more insightful of human learning, they are known to be significantly worse than backpropagation in performance evaluated by standard metrics (accuracy for classification, dice score for segmentation, mIoU for detection, etc.). In this paper, the authors aim to design a biologically plausible learning method that performs on par with backpropagation. Specifically, they propose pre-training with backpropagation and fine-tuning with biologically plausible Sign-Symmetry learning rules, and show that on image classification and image retrieval, the resulting models are more robust to perturbations from gradient-based adversarial attacks while performing comparably in standard metrics.

**Strengths:**

1.	The general topic of biologically plausible alternative to backpropagation is of interest to both machine learning and neuroscience communities.
2.	The background on adversarial attacks and adversarial robustness is helpful.
3.	The authors could further highlight the fact that they are directly tuning the publicly available models pre-trained on ImageNet, which means the proposed method is relatively efficient.

**Weaknesses:**

1.	I find the results presented quite alerting. **If I understand it correctly, in Table 1, the authors claimed that they could achieve 100.0% accuracy on ImageNet classification using backpropagation with VGG16 backbone.** I strongly doubt they are either presenting the train accuracy instead of the test accuracy or they are making other serious mistakes. I would like to encourage the authors to double check that and provide justification if my suspicion does not hold.
2.	The backbones being experimented are AlexNet, VGG16, and ResNet-18, and the first two are unfortunately no longer the mainstream backbones in recent years. I would recommend the authors provide rationale for choosing them rather than backbones such as Vision Transformer, ConvNeXT, etc.
3.	It would be especially helpful if the authors can go deeper on the insights on why training from scratch using Sign-Symmetry methods are subpar whereas finetuning using Sign-Symmetry methods from backpropagation-based pretrained weights are showing competitive performance. Insights into this phenomenon, especially if can be explained from a neuroscience perspective could be extremely interesting. With that said, everything has to rely on the fact that the results are reliable --- refer to Weakness 1.

**Questions:**

Please see questions 1 and 2. I would encourage the authors to directly respond to them, especially 1.

---

> ### Author Response · Authors · 2024-11-14
> **Questions Response**
>
> Response Q1:
>
> Thanks for pointing this out, indeed after submission we noticed that we had made a mistake that included training data in our test split (data contamination) for the case of ImageNet in image classification, however in Deep Hashing we have based our train/val/test on the same split used in the literature of Deep-Hashing (see HashNet/pytorch/data/imagenet/test.txt at master · thuml/HashNet - https://github.com/thuml/HashNet/blob/master/pytorch/data/imagenet/test.txt).
> We’re working on replacing ImageNet with NUS-WIDE and MSCOCO for the classification task, for the camera-ready version, this would avoid any confusion when it comes to the potential effect pre-training on ImageNet could have on our claim.
>
>
> Response Q2:
>
> The motivation behind using these backbones is that they already have established biologically plausible definitions for their respective architectures. In our approach, we propose a new method for training neural networks rather than a biologically plausible version of AlexNet or VGG-16 (work that has already been done). Adapting biological plausibility to new neural network architectures is another research topic, many architectures have no biologically plausible implementation (Transformers, for example).
> The results of our work motivate research on proposing biologically plausible versions of recent architectures (a topic that has yet to see much activity due to the performance problems we have mentioned).
>
>
>
> Response Q3:
>
> While we lack formal mathematical proofs or neuroscientific evidence, there are compelling theoretical arguments supporting this method's efficiency:
>
>  - Regarding robustness compared to backpropagation: Biologically plausible methods operate without complete weight information, which is typically required for gradient-based optimization attacks. The effect of using noised weights (brSF/frSF) or only the sign of the amplitude of the weights(uSF) “fools” gradient-based attacks, and makes them perform less effectively.
>
> - Concerning performance relative to BP: Our hypothesis suggests that while BP demonstrates superior capability in representation learning during the pre-training phase (which has already been shown in [1]), both BP and sign-symmetric methods achieve comparable results in task adaptation (transfer learning) during fine-tuning. Furthermore, biologically plausible methods exhibit an inherent advantage in robustness.
>
> [1] Assessing the Scalability of Biologically-Motivated Deep Learning Algorithms and Architectures - https://proceedings.neurips.cc/paper_files/paper/2018/file/63c3ddcc7b23daa1e42dc41f9a44a873-Paper.pdf

---

> > ### Comment · Reviewer_Xapq · 2024-11-24
> > **Response to rebuttal**
> >
> > I would thank the authors for their rebuttal.
> >
> > I am generally satisfied with Q2 and Q3. However, Q1 is still standing out as a major concern. While I trust the authors that it was unintentional, in my opinion it would be improper to accept this current version of the manuscript before this major technical error is fixed.

---

### Official Review · Reviewer_nQAk · 2024-11-03

**Soundness:** 3
**Presentation:** 2
**Contribution:** 2
**Rating:** 5
**Confidence:** 4

**Summary:**

The paper investigates an alternative training method for deep neural networks to address the robustness limitations of traditional backpropagation (BP). Specifically, it proposes a hybrid approach where neural networks are pre-trained with BP for initial accuracy and then fine-tuned using biologically plausible learning rules, particularly Sign-Symmetry methods (uSF, frSF, brSF). This fine-tuning method aims to enhance robustness against adversarial attacks without sacrificing the accuracy obtained through BP. The paper tests this method across multiple neural network architectures (AlexNet, VGG16, ResNet-18) on tasks including image classification and hashing-based image retrieval, demonstrating improved adversarial robustness while retaining accuracy levels comparable to standard BP training.

**Strengths:**

The approach of pre-training with BP and fine-tuning with bio-plausible learning rules is interesting and presents a promising path toward enhancing robustness in neural networks.

The experiments are well-designed, with comprehensive comparisons across different architectures, learning rules, and tasks.

The findings demonstrate that bio-plausible learning can play a critical role in improving adversarial robustness, which could inspire further research in bio-inspired ML methods.

**Weaknesses:**

The proposed method's robustness against adversarial attacks is limited to specific gradient-based attacks (FGSM and PGD). Testing on additional types of attacks, including non-gradient-based ones, would have helped in better understanding the limitation.

Some sections on bio-plausible learning rules and their relationship to robustness could benefit from a more detailed explanation to provide a clearer understanding of why these methods improve adversarial robustness.

While the chosen datasets are common benchmarks, including a wider range of datasets could enhance the robustness of claims’ generalizability.

**Questions:**

Can you elaborate on the choice of datasets used for testing? Were they selected based on specific characteristics that highlight the strengths of your approach?

How does your method perform in scenarios involving non-gradient-based adversarial attacks?

What are the potential limitations or drawbacks of using bio-plausible methods in other domains or applications?

---

> ### Author Response · Authors · 2024-11-14
> **Questions Response**
>
> Q1: Can you elaborate on the choice of datasets used for testing? Were they selected based on specific characteristics that highlight the strengths of your approach?
>
> Response: The choice of dataset was not constrained by any specific requirements, as our approach is designed to be generalizable across all datasets. While we tested our method on ImageNet (which was used for pre-training), we also evaluated it on other datasets including CIFAR10, NUS-WIDE, and MS-COCO. We are currently working on incorporating NUS-WIDE and MS-COCO for the classification task in the camera-ready version.
>
> Q2: How does your method perform in scenarios involving non-gradient-based adversarial attacks?
>
> Response: These types of methods were not tested due to the nature of these attacks, which don't rely on gradient knowledge. We argue that bio-plausible methods are inherently more robust to gradient-based attacks only. However, we agree that including these methods would help provide more insights into the behavior of such learning approaches.
>
> Q3: What are the potential limitations or drawbacks of using bio-plausible methods in other domains or applications?
>
> Response: previous works have only pointed out performance issues [1], To the best of our knowledge, we’re the first to apply sign-symmetry to other task than Image Classification, and we show that it performs relatively at the same scale as BP with a robustness advantage when used in the fine-tuning step.
>
> [1] Assessing the Scalability of Biologically-Motivated
>  Deep Learning Algorithms and Architectures -  https://proceedings.neurips.cc/paper_files/paper/2018/file/63c3ddcc7b23daa1e42dc41f9a44a873-Paper.pdf

---

> > ### Comment · Reviewer_nQAk · 2024-11-28
> > **response to rebuttal**
> >
> > Thanks for your explanation. I am unconvinced with the Q2 response, which needs to be addressed by incorporating the comparison with such adversarial attacks in the manuscript to raise the score.

---

### Official Review · Reviewer_YhiQ · 2024-11-06

**Soundness:** 2
**Presentation:** 2
**Contribution:** 2
**Rating:** 5
**Confidence:** 3

**Summary:**

The paper presents an approach combining traditional backpropagation for pre-training and bio-plausible learning rules, specifically Sign-Symmetry, for fine-tuning. The authors argue that this hybrid training method results in models that are more robust to gradient-based adversarial attacks compared to standard backpropagation-only models, while achieving similar accuracy and precision.

**Strengths:**

The experimental results convincingly demonstrate the approach’s increased robustness against adversarial attacks. This aligns with the authors’ intent to mitigate backpropagation's vulnerability, leveraging bio-plausible learning's inherent resilience.

The paper integrates backpropagation and bio-plausible methods, showing that such a hybrid model can preserve performance while enhancing security against adversarial examples. This is a promising area for further exploration in the community.

The experiments cover multiple backbone architectures, learning rules, and datasets, providing a comprehensive view of the proposed approach's effectiveness across different settings.

**Weaknesses:**

The motivation for pre-training on ImageNet and then fine-tuning on ImageNet100 could use clarification. It’s unclear why a smaller dataset subset was chosen for fine-tuning, as this could introduce a slight overfitting risk with backpropagation, affecting robustness. It would also be helpful to see comparisons with results before fine-tuning, particularly regarding robustness metrics.

The term "fine-tuning" might be more accurately described if it involves "training the downstream task with unfrozen backbone weights," as suggested by the unclear distinction between backpropagation and bio-plausible stages. Making this distinction explicit would clarify the contribution.

The work by Sanfiz & Akrout (2021), used as a basis for some assumptions, is not peer-reviewed. While this may not significantly affect the paper’s core findings, it would be beneficial to rely on more robustly vetted sources where possible.

Figure 1's visualization of weight transport methods is not intuitive. The arrow representation could be misleading, as it does not clearly illustrate the differences in weight transport between backpropagation and bio-plausible methods. A more structured schematic detailing learning rules and their weight transport levels would improve clarity.

Although the paper claims to open-source the code, no direct link or supplemental is provided. Including a repository link would make replication easier for the community.

Adding explicit units (e.g., [%] for accuracy) in the tables would improve clarity. While it’s often implicit, making this explicit could aid comprehension.

This paper presents a legitimate and straightforward approach with practical relevance. The hybrid training method appears effective, and the results suggest it’s a viable direction for increasing adversarial robustness in neural networks. Further refinements, particularly in clarifying the fine-tuning method and visualizations, would enhance the clarity and impact of this work.

**Questions:**

Could you clarify the choice of pre-training on ImageNet and fine-tuning on ImageNet100? This choice seems unconventional, as it might introduce overfitting concerns or limit the evaluation’s generalizability.

When you refer to "fine-tuning," are you adjusting all model weights, including the backbone, with bio-plausible methods? Or is this term meant to describe training for downstream tasks with specific layers?

Figure 1’s arrow representation of weight transport is somewhat ambiguous. What does the arrow signify in terms of the amount of weight transport across methods?

Could you confirm that accuracy and mean average precision are presented in percentage terms in the tables?

The robustness analysis includes standard gradient-based attacks like FGSM and PGD, but no specialized attacks for bio-plausible learning methods. How might these attacks perform differently if evaluated against attacks tailored specifically to the Sign-Symmetry methods?

Could you provide an alternative visualization comparing learning rules and weight transport? Perhaps a comparison matrix or schematic diagram that delineates key differences?

---

> ### Author Response · Authors · 2024-11-14
> **Questions Response**
>
> Q1: Could you clarify the choice of pre-training on ImageNet and fine-tuning on ImageNet100? This choice seems unconventional, as it might introduce overfitting concerns or limit the evaluation’s generalizability.
>
> Response: The choice of pre-training on ImageNet was primarily motivated by established practices in the Deep Hashing image retrieval literature (see Section 3.1.1 - Network Architecture) [1]. Similarly, our benchmarking on ImageNet100 followed precedents from this literature [2]. We acknowledge that for our claims, this approach might raise concerns about potential overfitting affecting robustness and performance. To address these concerns, we evaluated our method on additional datasets (such as CIFAR10) for classification tasks. Furthermore, to strengthen the generalizability of our claims, we considered image retrieval as an additional task, using two datasets independent of ImageNet (NUS-WIDE and MSCOCO). For the camera-ready version, we are working to replace the ImageNet classification benchmark with NUS-WIDE and MSCOCO to eliminate any potential confusion in our results.
>
> Q2: When you refer to "fine-tuning," are you adjusting all model weights, including the backbone, with bio-plausible methods? Or is this term meant to describe training for downstream tasks with specific layers?
>
> Response: All the model parameters are fine-tuned using the bio-plausible learning rule (see Method - Learning Method and Model Architectures section).
>
> Q3: Figure 1’s arrow representation of weight transport is somewhat ambiguous. What does the arrow signify in terms of the amount of weight transport across methods?
>
> Response: The arrow indicates the amount of weight information used in the backward pass. Backpropagation uses 100% of the weight information, thus being at the most extreme right. uSF uses only the sign of weights, thus using less information than BP, but more than brSF/frSF which uses the sign while adding some noise to the amplitude. FA (Feedback Alignment) doesn't use any weight information (0%).
>
> Q4: Could you confirm that accuracy and mean average precision are presented in percentage terms in the tables?
>
> Response: Yes we do confirm it’s in percentages, thanks for pointing this out, we’ll consider mentioning this to avoid any confusion.
>
> Q5: The robustness analysis includes standard gradient-based attacks like FGSM and PGD, but no specialized attacks for bio-plausible learning methods. How might these attacks perform differently if evaluated against attacks tailored specifically to the Sign-Symmetry methods?
>
> Response: Indeed, gradient-based attacks are not specifically tailored for bio-plausible learning rules. Therefore, we hypothesize that the adversarial robustness comes inherently from the noise added to the weights, as compared to exact methods like backpropagation (BP). The literature has not yet proposed any attacks specific to bio-plausible learning rules; this is an open research question that we mention in the discussion section (see the last sentence of the last paragraph).
>
> Q6: Could you provide an alternative visualization comparing learning rules and weight transport? Perhaps a comparison matrix or schematic diagram that delineates key differences?
>
> Response: Thanks for pointing this out, we are working on a more comprehensive figure, one that would show each matrix and the difference between each method in terms of weight transport, we’re also considering a more intuitive representation of the axis.
>
> [1] A Survey on DeepHashing Methods - https://dl.acm.org/doi/pdf/10.1145/3532624
>
> [2] HashNet: Deep Learning to Hash by Continuation - https://openaccess.thecvf.com/content_ICCV_2017/papers/Cao_HashNet_Deep_Learning_ICCV_2017_paper.pdf

---

### Meta-Review · Area_Chair_XVWF · 2024-12-18

**Metareview:**

This paper investigates bio-plausible learning rules for training neural networks due to their robustness to adversarial attacks. Prior work demonstrates this robustness but not accuracy. In this work, they address this by proposing sign-symmetry fine-tuning in addition to traditional backpropagation pre-training to enjoy both high accuracy as well as robustness to gradient-based attacks.

The paper's strengths are in its interesting contribution to hybrid training methods for robustness and accuracy which is a very promising research direction. However, the reviewers raised multiple concerns that were not completely clarified in the rebuttal: (a) dataset contamination with training data that may have contributed to some incorrect results, (b) limited evaluations to gradient-based attacks make the contributions less comprehensive, and (c) presentation details are not up to the mark, with one reviewer citing "ambiguous terminology, inadequate visualizations, and missing experimental details".

Due to the above concerns, I believe the paper cannot be accepted in its current form.

**Additional Comments On Reviewer Discussion:**

In addition to the above concerns, the reviewers also raised other questions regarding technical details that I believe the authors have satisfactorily addressed in the rebuttal.
The main concern with incorrect results that the authors may fix in its revision is still a major issue followed by presentation flaws and incremental contributions. All these issues can be fixed through the reviewer suggestions in future versions of the paper.

---

### Decision · Program_Chairs · 2025-01-22

Reject